# The Influence of Mg-Based Inclusions on the Grain Boundary Mobility of Austenite in SS400 Steel

**Chih-Ting Lai [1], Hsuan-Hao Lai [1], Yen-Hao Frank Su [2]**  **, Fei-Ya Huang [1], Chi-Kang Lin [1], Jui-Chao Kuo [1,*] and Hwa-Teng Lee [3]**

[1] Department of Materials Science and Engineering, National Cheng-Kung University, Tainan 701, Taiwan; tedchen1118@hotmail.com (C.-T.L.); lai57215308@gmail.com (H.-H.L.); a2780490@gmail.com (F.-Y.H.); cold19871025@gmail.com (C.-K.L.)
[2] China Steel Corporation, Kaohsiung 81233, Taiwan; 150151@mail.csc.com.tw
[3] Department of Mechanical Engineering, National Cheng-Kung University, Tainan 701, Taiwan; htlee@mail.ncku.edu.tw
[*] Correspondence: jckuo@mail.ncku.edu.tw; Tel.: +886-6-2754194

**Abstract:** In this study, the effects of the addition of Mg to the grain growth of austenite and the magnesium-based inclusions to mobility were investigated in SS400 steel at high temperatures. A high-temperature confocal scanning laser microscope (HT-CSLM) was employed to directly observe, in situ, the grain structure of austenite under 25 torr Ar at high temperatures. The grain size distribution of austenite showed the log-normal distribution. The results of the grain growth curves using 3D surface fitting showed that the $n$ and $Q$ values of the growth equation parameters ranged from 0.2 to 0.26 and from 405 kJ/mole to 752 kJ/mole, respectively, when adding 5.6–22 ppm of Mg. Increasing the temperature from 1150 to 1250 °C for 20 min and increasing the addition of Mg by 5.6, 11, and 22 ppm resulted in increases in the grain boundary velocity. The effects of solute drag and Zener pinning on grain boundary mobility were also calculated in this study.

**Keywords:** confocal scanning laser microscope; grain growth; magnesium addition; low-carbon steel; mobility; pinning effect

## 1. Introduction

Both medium and thick steel plates are used in the building of high-rise buildings, bridges, and in ships. A high-heat-input welding technique with heat input of over 50 kJ/cm has been employed for high-efficiency, low-cost fabrication of these steel plates. Increasing the heat input energy during welding enhances the coarsening of the austenite grains in the heat-affected zone (HAZ). However, coarse grains have detrimental effects on the impact toughness of weldments [1].

TiN inclusions were first applied to improve the toughness of HAZs, which serve as potential nucleation sites for acicular ferrite (AF) formation [2–4]. The development of TiN inclusions [5–8] can be expressed as the zero generation of inclusions for AF formation, before the use of oxide inclusions. However, Nb, V, and Ti nitrides will dissolve into austenite because of the high heat input from welding temperatures above 1300 °C [9]. Thus, Mizoguchi et al. [10] first proposed oxide inclusions to improve the impact toughness of HAZs under high-heat-input welding.

Titanium and magnesium oxides can be called the first and second generations of oxide inclusions for AF formation. In this study, Titanium oxides serve as potent nucleation sites for the formation of intragranular AF [11–19]. Magnesium oxide serves as fine dispersed inclusions in steel [20] because Mg has a strong affinity for oxygen and a reasonable amount of Mg is soluble in steel [5]. Kojima et al. [5] first proposed magnesium oxide metallurgy technology, in which MgO, MgS, and Mg(O,S)

inclusions were used to pin grain boundaries. According to previous research, Zener pinning was the influence of the dispersion of spherical particles [21], ellipsoidal particles [22] and cylindrical particles [23] on the movement of grain boundaries. The particles acted to prevent the motion of grain boundaries by exerting a pinning force which counteracted the driving force of the grain boundaries. The pinning efficiency against grain growth for grain coarsening was determined by the particle size [21]. This process refined the HAZ microstructure. Suito et al. [24–27] reported that MgO and soluble Mg can hinder excessive growth of austenite grains. In addition, Zhu et al. found that the addition of 0.005 wt% Mg can effectively inhibit austenite grain growth [28,29] and lead to the formation of AF [30].

The chemical composition of steel, the cooling rate at temperatures ranging from 800 to 500 °C, the size of austenite grains, and the inclusion parameters affect the formation of intragranular AF in metal [31]. In this study, we investigated the effect of the addition of Mg on austenite grain growth in SS400 low-carbon steel using a high-temperature confocal scanning laser microscope (HT-CSLM) in order to observe the high temperature microstructure in situ. In addition, the pinning effect of the magnesium-based inclusions on the mobility of austenite was studied.

## 2. Materials and Experiments

SS400 low-carbon steels produced by China Steel Corporation were selected as the experimental materials, for which the chemical composition of the steels is listed in Table 1, and Fe-5% Mg alloy wire was used for the addition of Mg during the ladle treatment. The experimental materials obtained from the as-cast slab, 8000 mm long, 1800 mm wide, and 270 mm thick, were cut into specimens 50 mm thick and 200 mm wide. After hot rolling at 1200 °C, the thickness was reduced from 50 mm to 10 mm, and 6 mm $\times$ 5 mm $\times$ 2 mm specimens were prepared.

**Table 1.** Chemical composition of SS400 steel with Mg of 1.5–22 ppm (Unit: wt%).

| Mg | C | Si | Mn | P | S | Al | N | O |
|---|---|---|---|---|---|---|---|---|
| 1.5 ppm | 0.120 | 0.17 | 1.18 | 0.0071 | 0.0006 | 0.028 | 0.0053 | 0.0041 |
| 5.6 ppm | 0.118 | 0.17 | 0.74 | 0.0102 | 0.0035 | 0.033 | 0.0054 | 0.0082 |
| 11.0 ppm | 0.136 | 0.28 | 0.94 | 0.0075 | 0.0012 | 0.020 | 0.0043 | 0.0027 |
| 22.0 ppm | 0.127 | 0.28 | 0.87 | 0.0104 | 0.0021 | 0.018 | 0.0052 | 0.0016 |

The specimens were ground using 1500, 2500, and 4000 SiC paper, polished using 3, 1, and 0.1 μm diamond suspensions, respectively, and finally polished with 0.02 μm $SiO_2$ suspensions. After the final polish, the austenite grain structure was observed in situ at high temperature under 25 torr Ar using an HT-CSLM (VL2000DX-SVF17SP, Yonekura, Japan). First, the specimen was placed into an $Al_2O_3$ crucible with a diameter of 8 mm and a height of 3.5 mm. Prior to heating, the chamber was evacuated for 5 min using a diffusion pump backfilled with ultrahigh purity Ar gas (>99.999%), with a discharge rate of 300 mL/min passed through a gas cleaning system to reduce the oxygen content. Then, the specimens were heated to 1000, 1100, 1200, and 1250 °C for 3, 5, 10, and 20 min, respectively. Afterward, the specimens were cooled at a cooling rate of 100 °C/s. After cooling, images of the grain structure were examined with a Leica DM6000 M optical microscope (Leica, Wetzlar, Germany). Microscope Imaging Software (Grain Expert, Leica, Wetzlar, Germany) was used to calculate the average grain size according to the ASTM E112 international standards. Then, the HT-CSLM was employed to investigate the movement of the austenite grains. We used an ASPEX EXplorer system (ASPEX, LLC, Boulder, CO, USA) combining SEM and EDS to automatically classify different types of inclusions and to determine the inclusion size and number within a large measured area.

## 3. Results and Discussion

### 3.1. Parameters of Grain Growth Equation

To determine the number of grains, we adopted the Leica analysis Grain Expert software to detect the grains, and Figure 1b shows an image of the detected grains in austenite. The original image is shown in Figure 1a. Then, a representative value of the minimal number of grains was obtained. Figure 2a–d show a histogram of grain size in terms of the number of grains, while Figure 2e shows the effect of the number of grains on the average grain size and standard deviation. The distributions of grain size and the average grain sizes were similar. However, increasing the number of grains decreased the standard deviation. The minimal number of grains was set at 300 for the subsequent analysis.

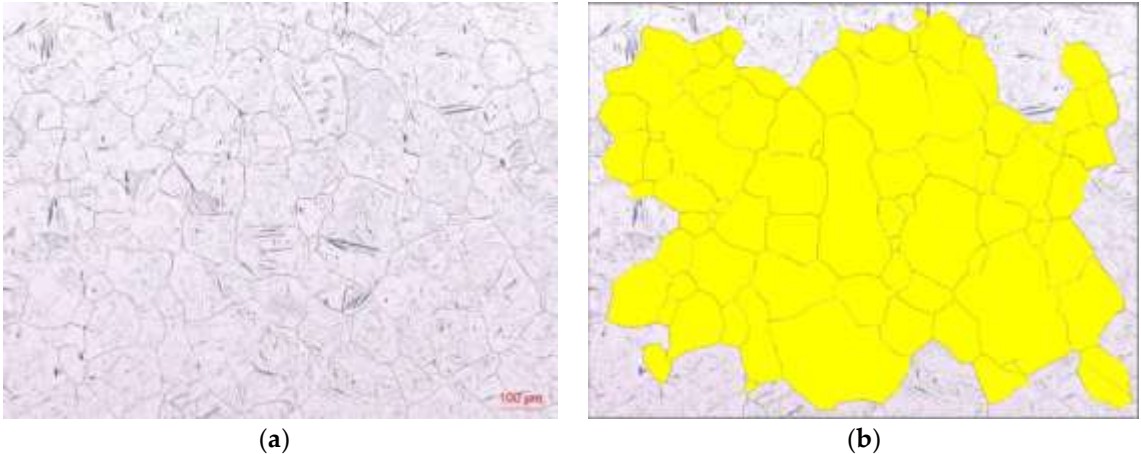

(**a**)        (**b**)

**Figure 1.** (**a**) Initial image of austenite obtained using optical microscopy and (**b**) after the detection process using analysis software after quenching.

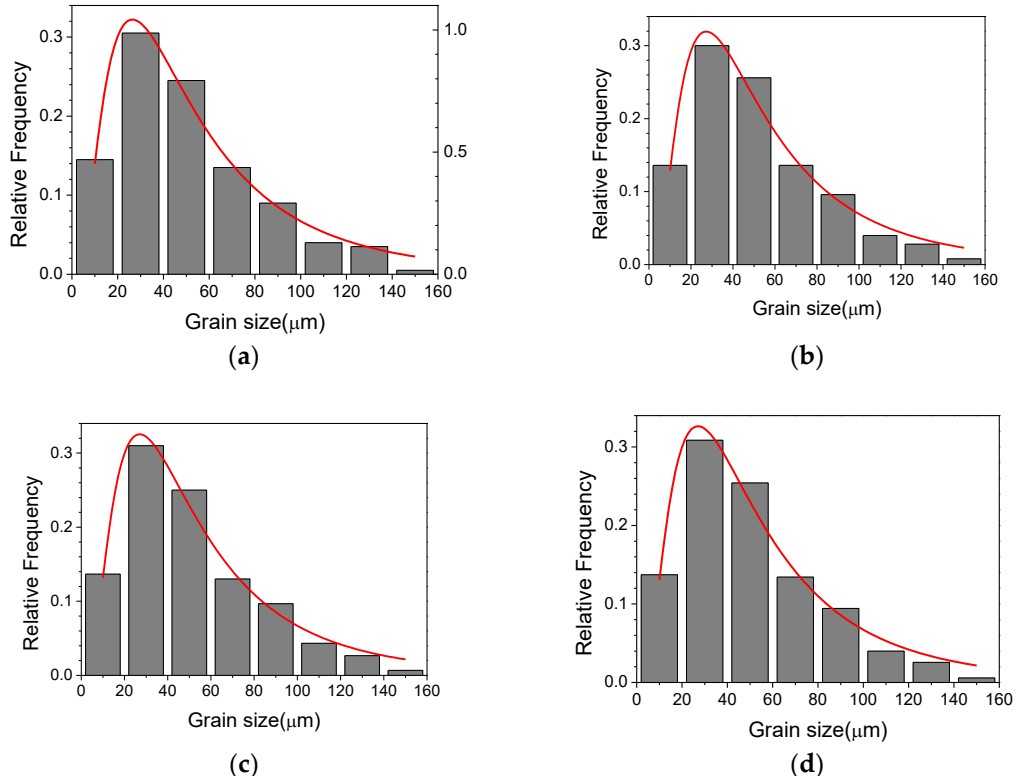

(**a**)        (**b**)

(**c**)        (**d**)

**Figure 2.** *Cont.*

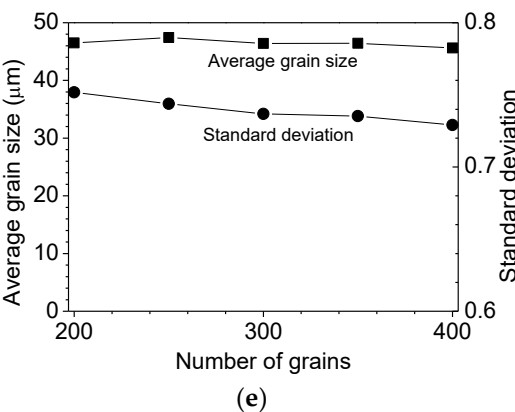

(**e**)

**Figure 2.** Histogram of austenite grain size in terms of the measured number of grains: (**a**) 200, (**b**) 250, (**c**) 300, and (**d**) 400. (**e**) is the average grain size and standard deviation as a function of the measured number of grains.

After measuring the number of grains, a single representative value for the grain sizes should be obtained statistically. Statistical values, including the mean, median, and mode, were considered. Figure 3 illustrates the histogram and cumulative frequency distributions of the grain size of austenite in SS400 with 1.5 ppm Mg. Figure 4 shows the evolution of the austenite grain size using the three statistical values. The mean is found by adding all given data and dividing by the number of data entries. The median is the middle number of all values, while the mode is the number that occurs most often in the data set. Thus, the mean has the highest value, while the mode is the lowest. The median is between the mean and mode (Figure 4). Therefore, in this study, we adapted the mode as the average value to avoid calculating the large grains because large grains lead to abnormal grain growth.

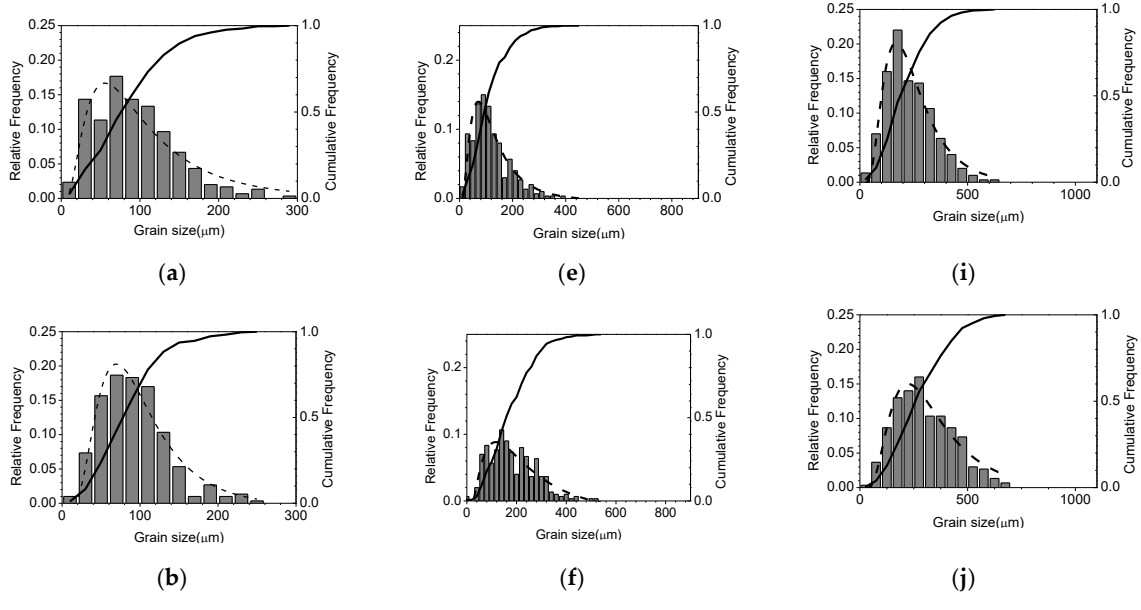

**Figure 3.** *Cont.*

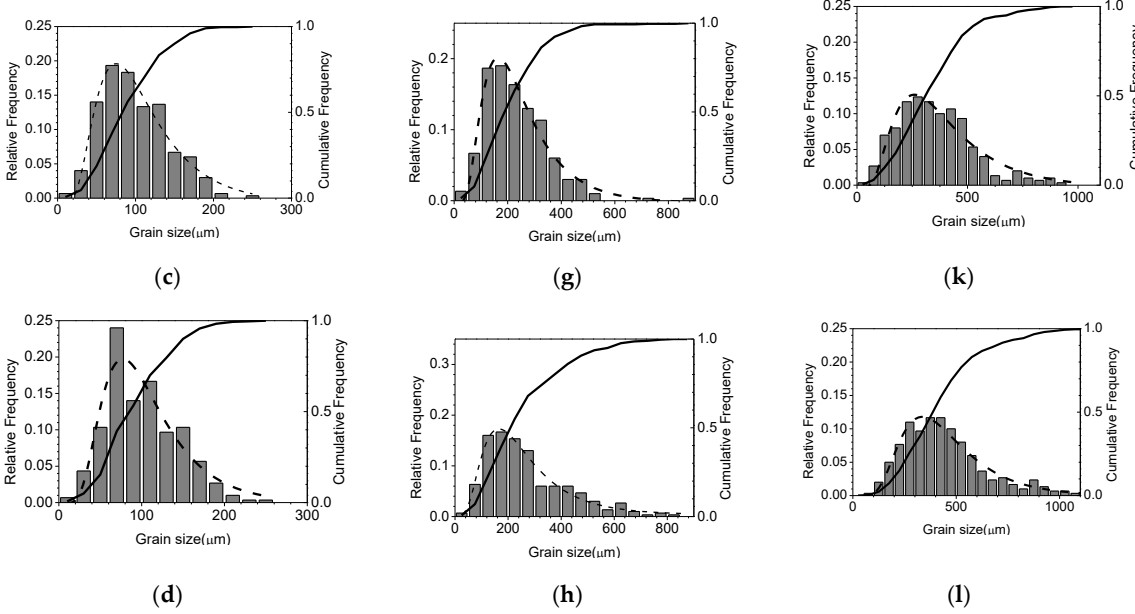

**Figure 3.** Histogram and cumulative frequency distributions of grain size of austenite in SS400 steel with 1.5 ppm Mg for (**a,e,i**) 3 min, (**b,f,j**) 5 min, (**c,g,k**) 10 min, and (**d,h,l**) 20 min. Broken lines represent the curves fitted by the log-normal function: (**a–d**) at 1150 °C, (**e–h**) at 1200 °C, and (**i–l**) at 1250 °C.

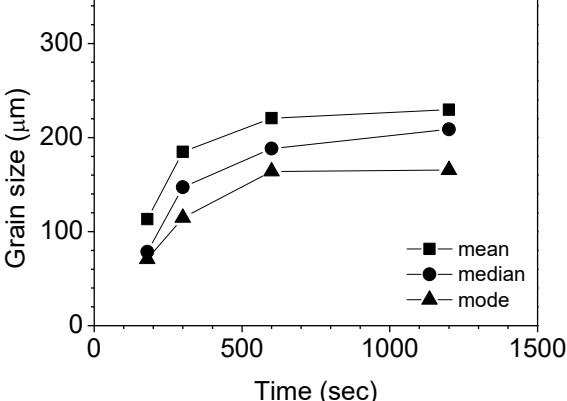

**Figure 4.** Grain size as a function of annealing time for SS400 with 1.5 ppm Mg at 1200 °C using the mean, median, and mode.

Next, the grain growth equation was derived after the averaging method was determined. The numerous equations that have been proposed to predict the grain growth equation can be divided into three types. First, Beck et al. [32] reported a power law for normal grain growth, while Sellars et al. [33–36] modified Beck's equation as follows:

$$d^m - d_o^m = k_0 \cdot \exp\left(-Q/RT\right) \cdot t \tag{1}$$

where *d* is the final grain diameter; $d_0$ is the initial grain diameter; *t* is the annealing time, and *m* and $k_0$ are constants. *Q* is the activation energy for grain growth, and *T* is the temperature in absolute degrees. The simplified Beck's model was used to predict the initial austenitic grain size as follows [37]:

$$d^m = k_0 \cdot \exp\left(-Q/RT\right) \cdot t \tag{2}$$

Moreover, Yoshie et al. [38] and Nishizawa [39] reported a model represented as follows:

$$d^2 - d_o^2 = k_0 \cdot \exp\left(-Q/RT\right) \cdot t \tag{3}$$

In this study, we applied the grain growth formulation in Equations (1) and (2) to predict the grain growth curves for SS400 with 1.5 ppm Mg at 1200 °C as an example. The parameters of the fitting growth equations are summarized in Table 2. The fitting curves, using Equation (2), agree with the experimental data with an R-value of 0.96, and the results are remarkably better than those using Equation (1) with an R-value of 0.69 [Figure 5a,b]. The values of $Q$ and $k_0$ using Equation (2) are similar to those of Equation (1). A comparison of the experimental data in Figure 5a,b showed that Equation (2) was the best fitting equation. Thus, Equation (2) was employed for the subsequent investigations.

**Table 2.** Parameters of the fitting equations of Equations (1) and (2) for SS400 with 1.5 ppm Mg.

| Fitting equation | $d_0$ (μm) | $m$ | $Q$ (kJ/mole) | $k_0$ (μm/s) |
|---|---|---|---|---|
| Equation (1) | 4.3 | 2 | 232 | $7.210^9$ |
| Equation (2) | 0 | 2.9 | 622 | $3.210^{25}$ |

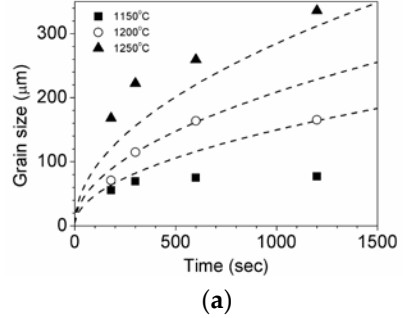 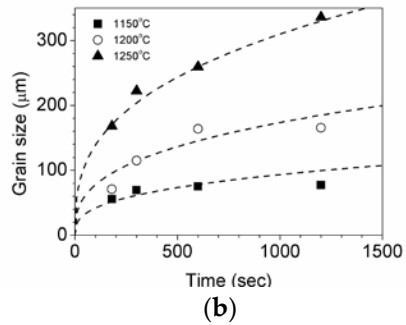

(**a**)         (**b**)

**Figure 5.** Mean value of grain size as a function of time using (**a**) Equation (1) with an R-value of 0.96 and (**b**) Equation (2) with an R-value of 0.69 for SS400 with 1.5 ppm Mg.

### 3.2. Effects of Temperature and Mg-Content on Austenite Grain Growth

Figure 3 illustrates the histogram and cumulative frequency distributions of the grain size of austenite with an Mg content of 1.5–22 ppm at 1150–1250 °C. The average grain size was plotted in terms of the annealing time and temperature with different Mg content, as shown in Figure 6, and the corresponding parameters of the fitting curves are listed in Table 3. The grain size distributions followed the log-normal distribution, which led to an increase in the average grain size with increases in annealing temperature. The $Q$ and n values are listed in Table 3.

**Table 3.** Parameters of the fitting equations for steel with Mg content of 1.5-22.0 ppm using Equation (2).

| Mg (ppm) | $m$ | $Q$ (kJ/mole) | $k_0$ (μm/s) |
|---|---|---|---|
| 1.5 | 2.9 | 622 | $3.2 \times 10^{25}$ |
| 5.6 | 4.9 | 405 | $2.5 \times 10^{19}$ |
| 11.0 | 3.8 | 493 | $2.8 \times 10^{21}$ |
| 22.0 | 6 | 752 | $1.7 \times 10^{31}$ |

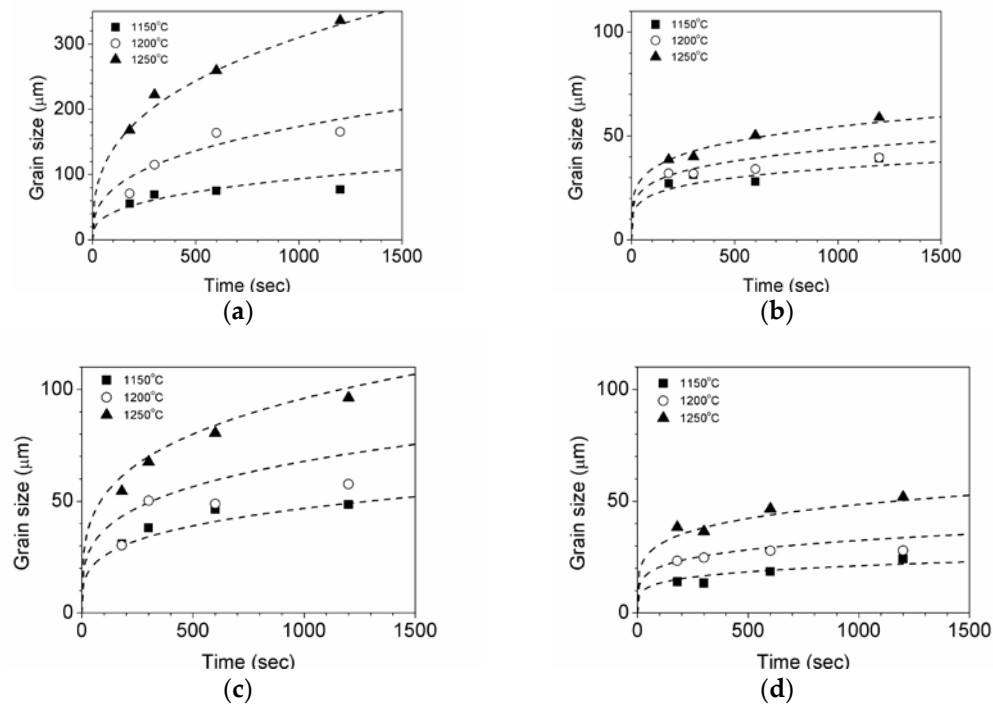

**Figure 6.** Grain size as a function of time for (**a**) 1.5, (**b**) 5.6, (**c**) 11, and (**d**) 22 ppm Mg, where symbols indicate experimental data and the fitting function is indicated by broken lines.

Except for the case of 1.5ppm Mg, the $Q$ value increased from 81 to 150 kJ/mole and the n value was in the range of 0.2–0.26 as Mg content increased from 5.6 to 22ppm. It was noted that the $Q$ value obtained from the references in Table 4 ranged from 64 to 437 kJ/mole with the exception of 914 kJ/mole. Thus, our $Q$ and *n* values lie in the range obtained from the references.

**Table 4.** Parameters of *n*, *Q*, and $k_0$ for the growth equations.

| Steels | *m* | Q (kJ/mol) | $k_0$ (µm/s) | References |
|---|---|---|---|---|
| Low C-Mn | 2 | 67 | $4.3 \times 10^{12}$ | [40] |
| Low C-Mn | 5.6 | 126 | $9.4 \times 10^{38}$ | [37] |
| C-Mn | 2 | 64 | $1.4 \times 10^{12}$ | [41] |
| C-Mn | 1 | 400 (T>1273) | $3.9 \times 10^{32}$ | [36] |
| C-Mn | 1 | 914 (T<1273) | $5.0 \times 10^{53}$ | [36] |
| C-Mn-V | 7 | 400 | $1.5 \times 10^{27}$ | [36] |
| C-Mn-Ti | 10 | 437 | $2.6 \times 10^{28}$ | [33] |
| C-Mn-Nb | 4.5 | 435 | $4.1 \times 10^{23}$ | [33] |
| High C-Mn | 5.3 | 366 | $9.1 \times 10^{25}$ | [37] |
| High C-Mn | 0.12 | 1170 | $4.1 \times 10^{63}$ | [31] |

We differentiated Equation (2) with respect to time (*t*) to reveal the grain boundary velocity ($V = \frac{dD}{dt}$) as follows:

$$V = n * k_o * \exp\left(-Q/RT\right) * [t - t_o]^{n-1} \tag{4}$$

The grain boundary velocity was plotted as a function of annealing time in Figure 7. When 1.5 ppm Mg was added, the grain boundary velocity increased from 0.025 µm/s to 0.1 µm/s after 20 min as the annealing temperature increased from 1150 to 1250 °C. The grain boundary velocity after 20 min increased from 0.005 µm/s to 0.01 µm/s, 0.01 µm/s to 0.02 µm/s, and 0.005 µm/s to 0.07 µm/s when 5.6, 11, and 22 ppm of Mg were added, respectively, as the annealing temperature increased from 1150 to 1250 °C. Considering the annealing time of 20 min, the grain boundary velocity was plotted in terms of Mg content at 1150–1250 °C in Figure 8. However, the grain boundary velocity was

reduced with increases in the Mg content, and steels with the addition of 5.6 ppm and 22 ppm of Mg exhibited similar trends. This phenomenon suggests that the grain boundary velocity was significantly decreased by increasing Mg content at 1250 °C. The results in Table 3 show that the increase in Mg content from 1.5 to 22 ppm increased the *Q* exponent. However, the *Q* exponent decreased when the Mg content was 5.6 and 11 ppm. In general, the activation energy *Q* for grain growth was affected by the amount and type of alloying elements. For materials with inclusion pinning and elements of solute drag, grain growth was difficult and activation energy increased. In some mechanisms, the *Q* exponent decreased. The m exponent depended on the grain growth mechanisms. When the m exponent was 2, the materials had no defects or inclusions. When the m exponent was 3, several phenomena, such as inclusions in the grain, were observed, which indicates the occurrence of the pinning effect. When the m exponent was 4 the alloy element exhibited diffusion in the grain boundaries and produced inclusion, which indicates the occurrence of pinning and solute drag effects [42]. Thus, the effects of the pinning force and solute drag on austenite grain mobility are considered in the subsequent section.

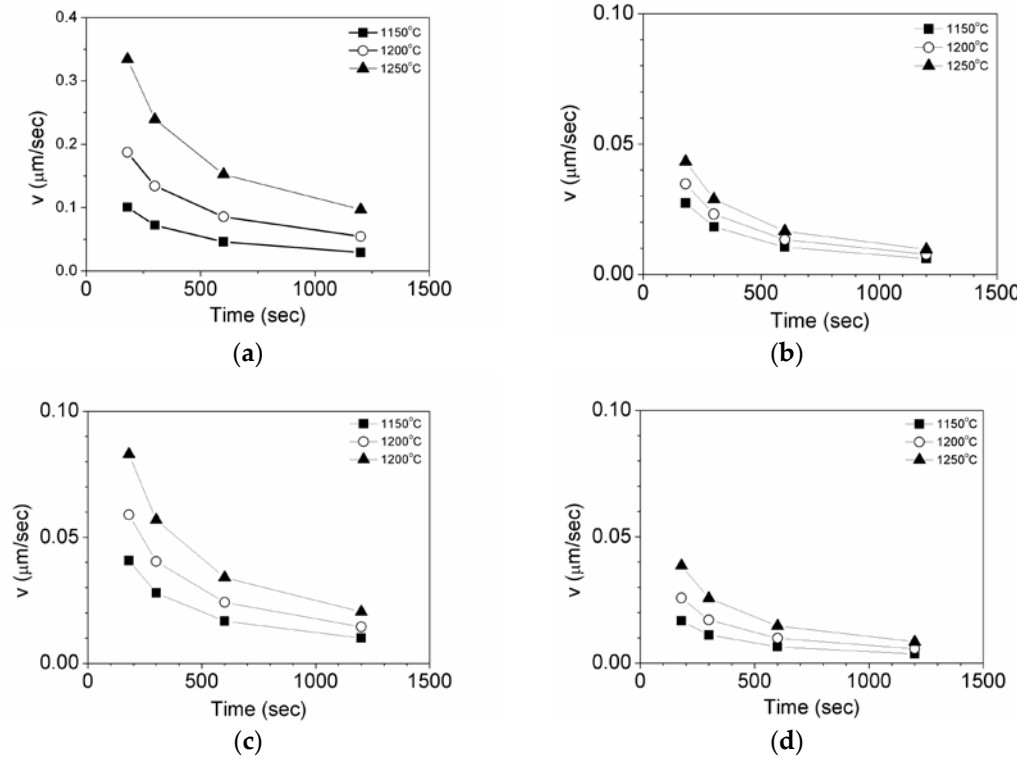

**Figure 7.** The grain boundary velocity (*V*) as a function of time for Mg at (**a**) 1.5, (**b**) 5.6, (**c**) 11, and (**d**) 22 ppm.

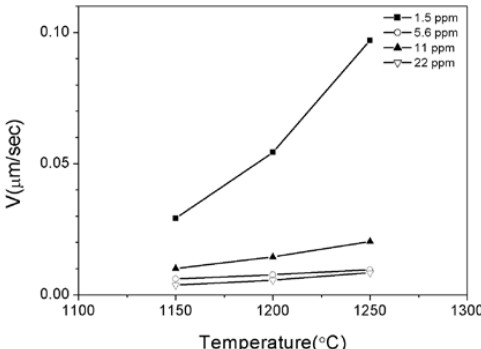

**Figure 8.** The grain boundary velocity (V) in terms of annealing temperature after 20 min of annealing time.

### 3.3. Zener Pinning Effect of Inclusions on the Austenite Grain Mobility

According to the model [36], the driving force of grain growth is expressed by:

$$F_P = \frac{2\sigma_b}{R} \tag{5}$$

where $R$ is the radius of the curvature of grains, and $\sigma_b$ is the surface energy of the grain boundary. Moreover, inclusions exert the retarding force ($F_R$) on the grain boundaries proposed by Zener [43] as:

$$F_R = \frac{3f_v\sigma_b}{2r} \tag{6}$$

where $f_v$ is the volume fraction of inclusions, $\sigma_b$ is the surface energy of the grain boundary, and $r$ is the radius of the inclusions. Here the surface energy of the grain boundary $\sigma_b$ is for austenite, the value of which was obtained from Reference [44]. The retarding forces (also called the Zener pinning force) depend on the radius and volume fraction of inclusions, as listed in Table 5. Here the radius ($r$) is equal to half of the average diameter, and $f_A$ is obtained by dividing the number of inclusions ($N$) by the measured area ($A$). Next, the equation $f_A = 2Nf_V$ was employed. It has been reported that the addition of 2 ppm Mg led to a change in oxide formation from the $Al_2O_3$ to $MgO \cdot Al_2O_3$ phase and a change in inclusion formation from $Al_2O_3$–MnS to $MgO \cdot Al_2O_3$–MnS [45].

**Table 5.** The retarding force ($F_R$) of inclusions.

| Mg (ppm) | Inclusion | $A$ (m²) | $D$ (m) | $f_A$ ($10^{-6} \times$ %) | $f_V$ ($10^{-4} \times$ %) | $F_R$ (kJ/m³) | $F_R$ (kJ/m³) |
|---|---|---|---|---|---|---|---|
| 1.5 | MgO-MnS | 87.6 | 1.9 | 2.5 | 15.9 | 2.9 | |
| | MnS | 67.9 | 1.4 | 2.0 | 14.0 | 3.5 | 11.7 |
| | MgAl₂O₄ | 693 | 2.9 | 19.9 | 44.6 | 5.3 | |
| 5.6 | MgO-MnS | 26.8 | 1.9 | 0.8 | 8.8 | 1.6 | |
| | MnS | 26.9 | 2.1 | 0.8 | 8.8 | 1.5 | 8.5 |
| | MgAl₂O₄ | 834 | 3.1 | 23.9 | 48.9 | 5.4 | |
| 11 | MgO | 10.4 | 1.7 | 0.2 | 4.7 | 1.0 | |
| | MgO-MnS | 46.6 | 1.9 | 1.0 | 10.0 | 1.8 | |
| | MnS | 861 | 3.4 | 18.5 | 43.0 | 4.3 | 20.3 |
| | MgAl₂O₄ | 2030 | 3.8 | 43.7 | 66.1 | 6.0 | |
| | Mg-Al-MnS | 2790 | 3.8 | 60.0 | 77.5 | 7.1 | |
| 22 | MgO | 108 | 2.7 | 3.1 | 17.6 | 2.2 | |
| | MgO-MnS | 224 | 4.4 | 6.4 | 25.3 | 2.0 | |
| | MnS | 310 | 2.5 | 8.9 | 29.8 | 4.2 | 17.1 |
| | MgAl₂O₄ | 235 | 2.8 | 6.7 | 26.0 | 3.2 | |
| | Mg-Al-MnS | 759 | 3.0 | 21.8 | 46.6 | 5.5 | |

* Note: $\sigma_b$ is 1.159 J/m² and the measured areas for 1.5, 5.6, 11, 22 ppm Mg are 34.865, 34.865, 46.487 and 34.865 mm², respectively. $A$—area; $D$—average diameter; $f_A$—area fraction; $f_V$—volume fraction; $F_R$—retarding force, and $F_R$—sum of the retarding forces for inclusions.

In the case of 1.5 and 5.6 ppm Mg, the effective inclusion of $MgAl_2O_4$ exerted the greatest retarding force, and, for 11 and 22 ppm Mg, it was Mg-Al-MnS that exerted the greatest retarding force because the volume fraction of $MgAl_2O_4$ was the largest phase for 1.5 and 5.6 ppm Mg. The volume fraction of Mg-Al-MnS was the largest phase for 11 and 22 ppm Mg. The average diameter of the inclusions was in the range of 1.4–4.4 μm, and the volume fraction of the inclusions was in the range of 4.7–77.5%. According to Equation (6), the retarding force was mainly determined by the volume fraction because there was no significant difference in the average diameter of inclusions seen in Table 5. Furthermore, there was no a linear relationship found between the addition of Mg and the retarding force.

The mobility of grain growth M is proportional to the net driving force, which is equal to the driving force of grain growth ($F_P$) subtracted from the Zener pinning force ($F_R$), which was expressed by Reference [43]:

$$V = M(F_P - F_R) \tag{7}$$

where $V$ is the grain boundary velocity. Figure 9 shows the grain boundary velocity in terms of the net driving force ($F_P - F_R$), where the grain boundary velocity was proportional to the net driving force. Here, the ratio of the grain boundary velocity to the net driving force corresponded to the mobility of grains M, that is, the slope, and the corresponding values of mobility are listed in Table 6. It was observed that increasing the temperature resulted in increases in mobility when 1.5–22 ppm of Mg were added because the mobility was dependent on the growth rate V also being proportional to $exp(-\frac{Q}{RT})$ in Equation (4). Here, we observed that an increase in Mg content led to a decrease in the mobility at 1150 °C as well as at 1200 and 1250 °C except for 5.6 ppm Mg content, as shown in Table 6. The grain boundary mobility was significantly reduced from 19.8 to $1.3 \times 10^{-4} \times$ m$^4$/(kJ·s) at 1150 °C, from 52.6 to $1.7 \times 10^{-4} \times$ m$^4$/(kJ·s) at 1200 °C, and from 105 to $3.7 \times 10^{-4} \times$ m$^4$/(kJ·s) at 1250 °C, when the Mg content was increased from 1.5 to 22 ppm.

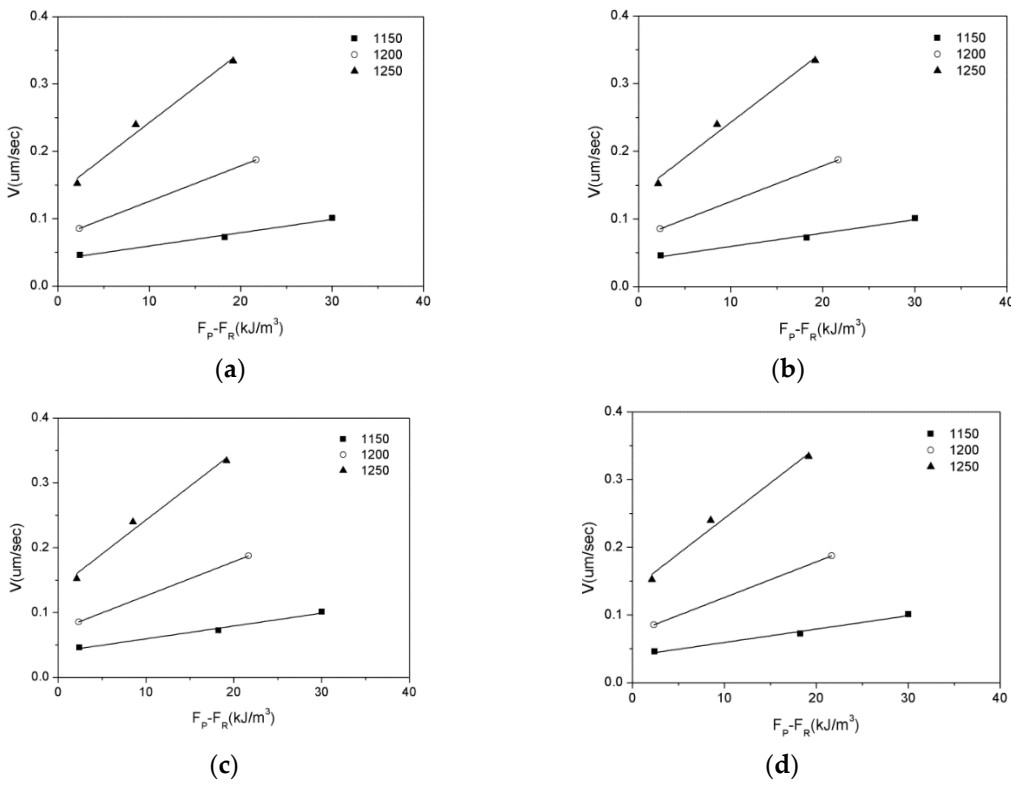

**Figure 9.** The grain boundary velocity ($V$) as a function of net driving force ($F_P - F_R$) for (**a**) 1.5 ppm, (**b**) 5.6 ppm, (**c**) 11 ppm and (**d**) 22 ppm Mg, where F$_P$ is the driving force of grain growth, and F$_R$ the retarding force on boundaries.

**Table 6.** The values of mobility [$10^{-4} \times$ m$^4$/(kJ·s)] for 1.5–22 ppm Mg at temperatures of 1150, 1200 and 1250 °C.

| Mg (ppm) | Temperature (°C) | | |
|---|---|---|---|
| | 1150 | 1200 | 1250 |
| 1.5 | 19.8 | 52.6 | 105 |
| 5.6 | 8.0 | 5.9 | 7.5 |
| 11.0 | 7.7 | 14.2 | 21.6 |
| 22.0 | 1.3 | 1.7 | 3.7 |

In this study, all measurements of grain size were based on a grooved surface. Grooving has a pinning effect on the migration of grain boundaries, which is therefore slow. The grain boundary mobility measured here was the lower limit of the grain boundary mobility.

### 3.4. Effect of Solute Drag on the Grain Mobility of Austenite

A lot of studies have reported that solute elements can segregate from the matrix to the grain boundary, and grain boundary migration may be limited by the diffusivity of the dragged solute atoms. The pressure induced by the solute drag is dependent on the boundary velocity [46–50]. According to Cahn et al., the drag effect of mobility (M) is given by [51]:

$$M(X_{Mg}, T) = \left( \frac{1}{M_{Pure}} + \alpha \cdot X_{Mg} \right)^{-1} \tag{8}$$

and

$$\alpha = \frac{N_v (kT)^2 \delta}{E_b D_{Mg}^{Int}} \left( sinh(\frac{E_b}{kT}) - (\frac{E_b}{kT}) \right) \tag{9}$$

where $M_{pure}$ is the mobility of grain boundary in Fe; $X_{Mg}$ is the concentration of Mg; $\delta$ is the width of a grain boundary; $N_v$ is the number of atoms per unit volume; $k$ is the Boltzmann's constant, and $E_b$ is the binding energy to the Fe grain boundary. The $M_{pure}$ is according to Kang et al. [52]:

$$M_{pure} = \beta \frac{D_{Fe}^{gb} V_m \delta}{b^2 RT} \tag{10}$$

where $b$ is the burger vector; $R$ is the gas constant; $T$ is the temperature; $D_{Fe}^{gb}$ is the diffusivity of Fe atoms in ferrite, and $V_m$ is the molar value of ferrite. The values of all the parameters used are listed in Table 7. The mobility calculation result was $1.8 \times 10^{-3} \times$ m$^4$/(kJ·s) at 1250 $^\circ$C for 22 ppm Mg. According to the pinning force results, the mobility of grain boundary was $3.4 \times 10^{-4} \times$ m$^4$/(kJ·s). Comparing the results, the effect of solute drag on grain boundary mobility is an order higher than the effect of Zener pinning. This implies that the effect of Zener pinning is the dominant mechanism for grain boundary migration.

**Table 7.** Values of parameters used in the calculation.

| Parameter | Value | Reference |
|:---:|:---:|:---:|
| $\delta$ | $1 \times 10^{-9}$ m | [53] |
| $b$ | $2.48 \times 10^{-10}$ m | [53] |
| $E_b$ | 85 kJ/mol | [54] |
| $\beta$ | 0.7 | [49] |
| $V_m$ | 8.26 x10$^{-5}$ m$^3$ | - |
| $1.8 \cdot 10^{-4}$ m$^2$/s | - | |
| $D_{Fe}^{g.b}$ | $1.5 \cdot 10^{-4}$ m$^2$/s | [55] |

## 4. Conclusions

In this study, the effect of Mg addition in the range of 1.5–22 ppm on the grain growth behavior of austenite in SS400 steel was investigated at 1150, 1200, and 1250 °C. The grain size distribution exhibited a log-normal distribution. The grain boundary velocity and grain mobility were reduced as Mg was increased from 1.5 ppm to 22 ppm. As the temperature increased from 1150 to 1250 °C, the grain boundary velocity also increased. The inclusion with the most retarding force was MgAl$_2$O$_4$, and it was Mg-Al-MnS when 11 and 22 ppm of Mg were added. The mobility of the grain boundary affected by Zener pinning was one order smaller than that affected by solute drag. It was therefore concluded that Zener pinning is a dominant factor by which to effectively retard grain boundary migration.

**Author Contributions:** Conceptualization, J.-C.K. and Y.-H.F.S.; methodology, C.-T.L.; software, H.-H.L.; validation, J.-C.K., Y.H. and H.-T.L.; formal analysis, H.-H.L.; investigation, Y.-H.F.S.; resources, H.-T.L.; data curation, C.-T.L.; writing—original draft preparation, H.-H.L.; writing—review and editing, C.-K.L.; supervision, J.-C.K.; project administration, F.-Y.H.; funding acquisition, Y.-H.F.S.

**Funding:** This research was funded by the Ministry of Science and Technology, grant number MOST 103-2622-E-006-037 and MOST 107-2221-E-006-018" and "The APC was funded by the Ministry of Science and Technology".

**Acknowledgments:** The authors would like to thank the Ministry of Science and Technology and China Steel Corporation for supporting the funding.

**Conflicts of Interest:** The authors declare no conflict of interest.

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
