# Peer review of "The Influence of Mg-Based Inclusions on the Grain Boundary Mobility of Austenite in SS400 Steel"

_metals, doi:10.3390/met9030370_

Reviewer 1 Report

The authors have made an attempt to investigate the pinning effect of micro-addition of magnesium in low carbon SS400 steel during heat treatment. Grain size was measured at different heat treatment conditions and some known models for grain growth were employed to fit the data. The effort by the authors in conducting these experiments are appreciable. However, some issues with the kinetic analysis, and interpretation of the data prevent me from recommending this manuscript for publication. Some issues that the authors might want to re-visit are given below

1.      Many confusing statements, uncorrelated texts and figures makes it hard to understand the context of the manuscript. For example, lines 91-92 reads “Figure 3 illustrates the histogram and cumulative frequency distributions of grain size of austenite in SS400 without Mg”. However, the figure represents something else as per caption of figure 3. Lines 125-126 refers back to Figure 3, where the text and figures correlate. Altogether, a lack of precision and clarity in presenting the data and text exists throughout the manuscript.

2.      The primary concern of authors in this manuscript is the effect of Mg inclusions on grain growth kinetics and they mention that the morphology and volume fractions of the inclusions are experimentally measured. However, no proof of this is shown in the manuscript. The only optical micrograph presented in Figure 1 doesn’t show any inclusion (at least in the magnification of image presented by the authors) although the caption for the figure reads just the opposite. 

3.      A lack of clarity on the derived activation energy (Q) exists. No description is given on how does the value of Q fit with physical processes such as grain boundary mobility when inclusions are present. The authors could make an attempt to relate the physical processes that drives the grain boundary mobility, when they have such good data. However, any discussion on this part is totally missing in the manuscript. The authors should at least make an effort to include some literature on how does the Q values (in the range they obtained) have been correlated with physical processes.

The parameter description of the Arrhenius relationships given by eq (1) and (2) do not match with that given in the tables either. For example, a fitting parameter ‘m’ was used in the equation, which is absent in the table. Another term ‘n’ was used in the tables, which was not described anywhere in the manuscript.

4.      The authors have repeatedly used a term ‘grain number’, which is confusing. In conventional metallography or EBSD, a grain number indicates an ID given to a particular grain. The text in section 3.1 and the corresponding figure (Figure 2) are not correlated. Please clarify if this is ‘grain number’ or ‘number of grains’?

5.      The authors need to check the language at many places of the manuscript. Understanding the context was very difficult due to incorrect grammar and a casual style of use of language. A perfect example for this scenario is lines 77 and 78. Such issues could be found throughout this manuscript. To convey the message unambiguously, the authors should consider re-writing this paper with more attention given to the language.

Author Response

Comment 1: Many confusing statements, uncorrelated texts and figures makes it hard to understand the context of the manuscript. For example, lines 91-92 reads “Figure 3 illustrates the histogram and cumulative frequency distributions of grain size of austenite in SS400 without Mg”. However, the figure represents something else as per caption of figure 3. Lines 125-126 refers back to Figure 3, where the text and figures correlate. Altogether, a lack of precision and clarity in presenting the data and text exists throughout the manuscript.

Respond:

Thank you for your suggestion, we modify to the current one.

Original

L91

Figure 3 illustrates the histogram and   cumulative frequency distributions of grain size of austenite in SS400   without Mg

The caption of Figure   3

Figure 3. Histogram and cumulative frequency   distributions of grain size of austenite in SS400 steel without Mg for (a, e,   i) 3 min, (b, f, j) 5 min, (c, g, k) 10 min, and (d, h, l) 20 min. Broken   lines represent the curves fitted by the log-normal function: (a–d) at   1150°C, (e–h) at 1200°C, and (i–l) at 1250°C.

L114

In this study, we applied the grain growth formulation   in Eqs. (1) and (2) to predict the grain growth curves for SS400 without Mg   at 1200 °C as an example.

The caption of Table   2

Parameters of the fitting equations of   Eq.(1) and (2) for SS400 without Mg.

Revised

L98

Figure 3 illustrates the histogram and cumulative   frequency distributions of grain size of austenite in SS400 with 1.5 ppm Mg

The caption of Figure   3

Figure 3. Histogram and cumulative frequency   distributions of grain size of austenite in SS400 steel with 1.5ppm Mg for   (a, e, i) 3 min, (b, f, j) 5 min, (c, g, k) 10 min, and (d, h, l) 20 min.   Broken lines represent the curves fitted by the log-normal function: (a–d) at   1150°C, (e–h) at 1200°C, and (i–l) at 1250°C.

L125

In this study, we applied the grain growth formulation   in eqs. (1) and (2) to predict the grain growth curves for SS400 with 1.5 ppm   Mg at 1200 °C as an example.

The caption of Table   2

Parameters of the fitting equations of   Eq.(1) and (2) for SS400 with 1.5 ppm Mg.

Comment 2: The primary concern of authors in this manuscript is the effect of Mg inclusions on grain growth kinetics and they mention that the morphology and volume fractions of the inclusions are experimentally measured. However, no proof of this is shown in the manuscript. The only optical micrograph presented in Figure 1 doesn’t show any inclusion (at least in the magnification of image presented by the authors) although the caption for the figure reads just the opposite.

Respond:

        We consider the volume fractions of the inclusions from the macro analysis. In this study, we only focus on different Mg concentration has the different pinning force, grain boundary velocity and mobility of grain boundary.

Comment 3: A lack of clarity on the derived activation energy (Q) exists. No description is given on how does the value of Q fit with physical processes such as grain boundary mobility when inclusions are present. The authors could make an attempt to relate the physical processes that drives the grain boundary mobility, when they have such good data. However, any discussion on this part is totally missing in the manuscript. The authors should at least make an effort to include some literature on how does the Q values (in the range they obtained) have been correlated with physical processes.

The parameter description of the Arrhenius relationships given by eq (1) and (2) do not match with that given in the tables either. For example, a fitting parameter ‘m’ was used in the equation, which is absent in the table. Another term ‘n’ was used in the tables, which was not described anywhere in the manuscript.

Respond:

In this study, we only discuss the macroscopic result of grain growth equation. But, addition different Mg concentration into steel will form the different number of inclusion type. Different inclusion type has different pinning force from my other research. It will affect the Q value. But we considered your suggestion in the next study. And the term “n” is modify to correct one.

Comment 4: The authors have repeatedly used a term ‘grain number’, which is confusing. In conventional metallography or EBSD, a grain number indicates an ID given to a particular grain. The text in section 3.1 and the corresponding figure (Figure 2) are not correlated. Please clarify if this is ‘grain number’ or ‘number of grains’?

Respond:

The term of “grain number” in this paper has been changed to “the number of grain” to describe the number of grain. Thank you for your suggestion.

Comment 5: The authors need to check the language at many places of the manuscript. Understanding the context was very difficult due to incorrect grammar and a casual style of use of language. A perfect example for this scenario is lines 77 and 78. Such issues could be found throughout this manuscript. To convey the message unambiguously, the authors should consider re-writing this paper with more attention given to the language.

Respond:

Thank for your suggestion, we have a ready sent the manuscript to foreign language center of university for revise.

Reviewer 2 Report

Title:

Should be modified. (Maybe: An influence of Mg-based inclusions on…)

Section „ Materials and Experimental”

Rows 58-61

What sections were prepared? longitudinal or transverse (short transverse or long transverse)? It should be given here.

Row 70

The procedure of the measurement of the velocity of the grain boundaries movement was not presented clearly.

Row 71

What was achieved using ASPEX Explorer system? The procedure should be described or given the literature reference.

In my opinion, essential for this paper is the definition of the “grain size”. It should be explained clearly in the text, what the “grain size” means here in the paper and how it was calculated (or the reference should be given).

In the equation (1) (row 107), the term “grain diameter” is applied. What is the correlation between the “grain size” used by the authors and the “grain diameter”. Moreover, what is the correlation between the used by the authors “grain size” with ASTM (E112 – 13) Grain Size Number?

Section „Conclusions)

Row 232

(For 1.5 and 5.6 ppm Mg…) – this conclusion does not have support in the results presented in this paper (maybe it is based on the other publication of the authors?)

Author Response

Comment 1: Title

Should be modified. (Maybe: An influence of Mg-based inclusions on…)

Respond:

According to your comment, the title is modify to The Influence of Mg- based Inclusion on the Mobility of Austenite in SS400 Steel

Comment 2: Rows 58-61

What sections were prepared? longitudinal or transverse (short transverse or long transverse)? It should be given here.

Respond:

The 5 cm thickness of as-cast slabs was taken along with the casting direction and cut into the size with a width of 20 cm. And then used hot rolling into the size with the thickness of 0.1 cm. We choose the red region to use as the specimens. The schematic illustration was showed as follow.

Comment 3: Row 70

The procedure of the measurement of the velocity of the grain boundaries movement was not presented clearly.

Respond:

In this study, the velocity of grain boundary was used equation to calculate the value. The velocity of the equation of grain boundaries () as follows :

Original

Revised

L155

We     differentiated Eq. (2) with respect to time t to reveal the grain     boundary velocity () as follows: 

Comment 4: Row 71

What was achieved using ASPEX Explorer system? The procedure should be described or given the literature reference.

Respond:

ASPEX EXplorer system combining SEM and EDS to automatically classify different types of inclusions and to determine the inclusion size and number within a large measured area. Thus, we can got the value of fA and fv. And then, FR was calculated by Zener equation. And we also write the procedure in the text.

Original

Revised

L75

We used an   ASPEX EXplorer system combining SEM and EDS to automatically classify   different types of inclusions and to determine the inclusion size and number   within a large measured area.

Comment 5:

In my opinion, essential for this paper is the definition of the “grain size”. It should be explained clearly in the text, what the “grain size” means here in the paper and how it was calculated (or the reference should be given).

In the equation (1) (row 107), the term “grain diameter” is applied. What is the correlation between the “grain size” used by the authors and the “grain diameter”. Moreover, what is the correlation between the used by the authors “grain size” with ASTM (E112 – 13) Grain Size Number?

Respond:

In my study, OM software (grain expert) was used to calculate the grain size. This software can use different international standards (ASTM, JIS, ISO etc) and we chose ASTM E112 to calculate the grain size. And we also explained clearly in the text. And the term of grain diameter of equation 1 and grain size are the same. The grain size number (n) is the number of grins from ASTM E112-13. In my study, the grain diameter is the grain size (d).

Original

Revised

L72

The OM software (Grain Expert) was used to   calculate the average grain size according to the ASTM E112 international   standards.

Comment 6: conclusions, Row 232

For 1.5 and 5.6 ppm Mg…) – this conclusion does not have support in the results presented in this paper (maybe it is based on the other publication of the authors?)

Respond:

According to your comment, we rewrite the conclusion.

Original

In this study, the effect of Mg   addition in the range of 1.5–22 ppm on the grain growth behavior of austenite   in SS400 steel was investigated at 1150 °C, 1200 °C, and 1250 °C. The grain   size distribution revealed a log-normal distribution. The results obtained   from 3D surface fitting showed that the n value ranged from 0.2 to   0.26, and the Q value ranged from 405 kJ/mole to 752 kJ/mole when   5.6–22 ppm of Mg were added. Adding 5.6, 11, and 22 ppm of Mg after 20 min   resulted in increasing the grain boundary velocity from 0.005 μm/s to 0.01 μm/s,   0.01 μm/s to 0.02 μm/s, and 0.005 μm/s to 0.07 μm/s, respectively,   as the temperature was increased from 1150 °C to 1250   °C. The grain boundary mobility was significantly reduced from 19.8 to 1.310-4m4/(kJ∙s)   at 1150   oC,   from 52.6 to 1.710-4m4/(kJ∙s)   at 1200   oC   and from 105 to 3.710-4m4/(kJ∙s)   at 1250   oC,   as increasing Mg from 1.5 to 22 ppm. For 1.5 and 5.6 ppm Mg addition, the   effective inclusion having the most retarding force was MgAl2O4,   and it was Mg-Al-MnS for 11 and 22 ppm Mg addition. The mobility of grain   boundary affected by Zener pinning is one order smaller than that affected by   solute drag. It’s concluded that Zener pinning is a dominant factor to   effectively retard grain boundary migration.

Revised

conclusions

In this study, the effect of Mg addition in the   range of 1.5–22 ppm on the grain growth behavior of austenite in SS400 steel   was investigated at 1150°C, 1200°C, and 1250°C. The grain size distribution   exhibited a log-normal distribution. The grain boundary velocity and grain   mobility were reduced as Mg was increased from 1.5 ppm to 22 ppm. As the   temperature was increased from 1150°C to 1250°C, the grain boundary velocity   also increased. The inclusion with the most retarding force was MgAl2O4,   and it was Mg-Al-MnS when 11 and 22 ppm of Mg was added. The mobility of the   grain boundary affected by Zener pinning was one order smaller than that   affected by solute drag. It was therefore concluded that Zener pinning is a   dominant factor by which to effectively retard grain boundary migration.

Reviewer 3 Report

The authors investigated the effect of Mg addition in the range of 1.5–22 ppm on the grain growth behavior of austenite in SS400 steel at 1150 °C, 1200 °C, and 1250 °C.

1. L115-117 : The authors mentioned that the fitting curves using Eq. (2) agree with the experimental data with R of 0.96, and the results are remarkably better than those using Eq. (1) R of 0.69 [Figures 5(a) and (b)]. However, in Figure 5 (a), the fitted lines at 1150  °C and 1250  °C were strayed out from the results at both temperatures. Please mention the reason of these differences. 

2. Table  3 : The Q value was not changed monotonically with increasing Mg (211→ 81→128→150). Please explain about this reason.

3. Figure 9 (a)-(d) : The fitting lines were not crossed across the  origin points (0, 0). Please explain about this reason.

4. Change all variables in Italic style, 

Author Response

Comment 1: L115-117 : The authors mentioned that the fitting curves using Eq. (2) agree with the experimental data with R of 0.96, and the results are remarkably better than those using Eq. (1) R of 0.69 [Figures 5(a) and (b)]. However, in Figure 5 (a), the fitted lines at 1150 °C and 1250 °C were strayed out from the results at both temperatures. Please mention the reason of these differences.

Respond:

It’s because the Eq. 1 has consider the initial grain size, Eq. 2 has not. It’s same as the stress -strain curve. If we only consider the plastic curve, the stress-strain curve will more accurate than consider the elastic curve and plastic curve.

Comment 2: Table 3 : The Q value was not changed monotonically with increasing Mg (211→ 81→128→150). Please explain about this reason.

Respond:

In this study, we only discuss the macroscopic result of grain growth equation. But, addition different Mg concentration into steel will form the different number of inclusion type. Different inclusion type has different pinning force from my other research. It will affect the Q value.

Comment 3: Figure 9 (a)-(d): The fitting lines were not crossed across the origin points (0, 0). Please explain about this reason.

Respond:

In general, the fitting lines were crossed across the origin points. In our fitting result shows that the lines were not crossed across the origin points due to the effect of other retarding force.

Comment 4: Change all variables in Italic style,

Respond:

According to your comment, all variables are changed to Italic style. 

Reviewer 4 Report

This paper describes an experimental study of the use of Mg to understand the mobility of austenite in a stainless steel. The experimental work and data analysis are competently done. There are some issues to address before acceptance.

Line 36, I think they mean titanium, not titan.

Error bars need to be added to all of the appropriate figures: 2, 3, 4, 5, 6, 7, 8, and 9 as well as to any appropriate tables.

What are the units for the pre-exponential factor k0? Length/time? Length raised to some power/time? Add the units.

There appears to be an error in the equations or in the tables. Is it m (equations) or n (tables)? Be consistent.

Why is there such a large variation in k0?

In section 3.2 first paragraph and elsewhere in the paper. If numbers are in the tables, do not repeat them in the text. Discuss the numbers but do not repeat them.

The entire conclusions section needs to be rewritten. Currently it is essentially a list of numbers. Change it to a set of conclusions on what has happened physically. They may want to add in a short discussion section as well to summarize their nice results and provide more physical context to them.

The paper needs revision before acceptance mostly to improve the quality of the presentation of the results.

Author Response

Comment 1: Line 36, I think they mean titanium, not titan.

Respond:

According to your comment, the problem of page 11 has been modified to correct one.

Original

Titan   and magnesium oxides can be called the first- and second-generations of oxide   inclusions for AF formation

Revised

L38

Titanium   and magnesium oxides can be called the first- and second-generations of oxide   inclusions for AF formation

Comment 2: Error bars need to be added to all of the appropriate figures: 2, 3, 4, 5, 6, 7, 8, and 9 as well as to any appropriate tables.

Respond:

If we add the error bars in the figures. It will make the figure much more complicated. Thus, it will be difficult for people to clearly understand the information in the figures.

Comment 3: What are the units for the pre-exponential factor k0? Length/time? Length raised to some power/time? Add the units.

Respond:

The unit of the pre-exponential factor k0 is length/ time. And it’s also add in the text.

Original

Revised

Table 2

Fitting equationd0 (mm)mQ (kJ/mole)k0 (μm/s)Eq.(1)4.322327.2                                                                                                                       109Eq.(2)02.96223.21025

Revised

Table 3

Mg (ppm)mQ (kJ/mole)k0 (μm/s)1.52.96223.210255.64.94052.5101911.03.84932.8102122.067521.71031

Revised

Table 4

Steelsm Q     (kJ/mol)k0 (μm/s)ReferencesLow C-Mn2674.31012[38]Low C-Mn5.61269.41038[39]C-Mn2641.41012[40]C-MnC-Mn11400 (T>1273)914 (T<1273)3.910325.01053[41][41]C-Mn-V74001.51027[41]C-Mn-Ti104372.61028[31]C-Mn-Nb4.54354.11023[31]High C-Mn5.33669.11025[39]High C-Mn0.1211704.1 x1063[39]

Comment 4: There appears to be an error in the equations or in the tables. Is it m (equations) or n (tables)? Be consistent.

Respond:

Thank you for your suggestion, we modify to consistent.

Comment 5: Why is there such a large variation in k0?

Respond:

In this study, we only discuss the macroscopic result of grain growth equation. But, addition different Mg concentration into steel will form the different number of inclusion type. Different inclusion type has different pinning force from my other research. It will affect the k0 value.

Comment 6: In section 3.2 first paragraph and elsewhere in the paper. If numbers are in the tables, do not repeat them in the text. Discuss the numbers but do not repeat them.

Respond:

Thank you for your suggestion.

Original

The Q   values of 211, 81, 128, and 150 kJ/mole were obtained when 1.5, 5.6, 11, and 22 ppm of Mg were   added, respectively (Table 3). The n   values ranged from 0.34 to 0.2 when Mg was added from 1.5 to 22ppm (Table 3).  

Revised

L143

The Q   and n values are listed in Table 3.

Comment 7: The entire conclusions section needs to be rewritten.

Respond:

According to your comment to rewritten the conclusions.

Original

In this study, the effect of Mg   addition in the range of 1.5–22 ppm on the grain growth behavior of austenite   in SS400 steel was investigated at 1150 °C, 1200 °C, and 1250 °C. The grain   size distribution revealed a log-normal distribution. The results obtained   from 3D surface fitting showed that the n value ranged from 0.2 to   0.26, and the Q value ranged from 405 kJ/mole to 752 kJ/mole when   5.6–22 ppm of Mg were added. Adding 5.6, 11, and 22 ppm of Mg after 20 min   resulted in increasing the grain boundary velocity from 0.005 μm/s to 0.01 μm/s,   0.01 μm/s to 0.02 μm/s, and 0.005 μm/s to 0.07 μm/s, respectively,   as the temperature was increased from 1150 °C to 1250   °C. The grain boundary mobility was significantly reduced from 19.8 to 1.310-4m4/(kJ∙s)   at 1150   oC,   from 52.6 to 1.710-4m4/(kJ∙s)   at 1200   oC   and from 105 to 3.710-4m4/(kJ∙s)   at 1250   oC,   as increasing Mg from 1.5 to 22 ppm. For 1.5 and 5.6 ppm Mg addition, the   effective inclusion having the most retarding force was MgAl2O4,   and it was Mg-Al-MnS for 11 and 22 ppm Mg addition. The mobility of grain   boundary affected by Zener pinning is one order smaller than that affected by   solute drag. It’s concluded that Zener pinning is a dominant factor to   effectively retard grain boundary migration.

Revised

Conclusions

In this study, the effect of Mg addition in the   range of 1.5–22 ppm on the grain growth behavior of austenite in SS400 steel   was investigated at 1150 °C, 1200 °C, and 1250 °C. The grain size   distribution exhibited a log-normal distribution. The grain boundary velocity   and grain mobility were reduced as Mg was increased from 1.5 ppm to 22 ppm. As   the temperature was increased from 1150 °C to 1250 °C, the grain boundary   velocity also increased. The inclusion with the most retarding force was   MgAl2O4, and it was Mg-Al-MnS when 11 and 22 ppm of Mg was added. The   mobility of the grain boundary affected by Zener pinning was one order   smaller than that affected by solute drag. It was therefore concluded that   Zener pinning is a dominant factor by which to effectively retard grain   boundary migration.

Reviewer 5 Report

The present manuscript addresses an interesting topic on influence of Mg inclusions on mobility of austenitic grains. Overall, the manuscript is appropriately organized, but research and presentation lack at accuracy. The text is understandable to read, but, nevertheless, an extensive editing of English language and style is recommended.

Main question is, if microstructure of investigated steel is 100% austenitic. Figure 1 shows untypical substructures of grains, the microstructure is more similar to martensite. Due to the lattice-like substructures of the grain, the detection of grain boundary is inaccurate. Further, the description of Figure 1 (b) refer to “inclusion shape”, which is clearly not that is shown in the image. At the following text (lines 96-97), authors mention that large grains were not considered in the calculation, which is not understandable from scientific point of view (Fig. 1 shows big amount of grains larger than 160 µm, which are clearly not considered in Fig. 2). The histograms of grain distribution depicted in Fig. 2 and Fig.3 show different amount of grain size fractions, which is clearly make impact on fitting function.

In general, the article has serious flaws, additional experiments needed, research not conducted correctly. While a promising study was presented, the research design and the subsequent interpretation of results do not reach an appropriate scientific level.

Author Response

Reviewer 5

Comment 1: Main question is, if microstructure of investigated steel is 100% austenitic. Figure 1 shows untypical substructures of grains, the microstructure is more similar to martensite. Due to the lattice-like substructures of the grain, the detection of grain boundary is inaccurate.

Respond:

The OM software will use the gray scale to determine the austenite grain boundary or martensite. And, the detection of grain boundary was calculated on the austenite grain boundary as following figure. We don't detect intergranular martensitic.

Comment 2: Further, the description of Figure 1 (b) refer to “inclusion shape”, which is clearly not that is shown in the image. At the following text (lines 96-97), authors mention that large grains were not considered in the calculation, which is not understandable from scientific point of view (Fig. 1 shows big amount of grains larger than 160 µm, which are clearly not considered in Fig. 2). The histograms of grain distribution depicted in Fig. 2 and Fig.3 show different amount of grain size fractions, which is clearly make impact on fitting function.

Respond:

Thank you for your suggestion, the caption of Figure 1 (b) is modify to correct. Figure 1 is not the same condition with Figure 2. Figure 1 just want to show the how to determine the grain size by OM software in this study. Because large grains lead to abnormal grain growth. Thus, we only consider the grain size of normal grain growth. 

Round  2

Reviewer 1 Report

The effort by authors in improving the quality of the manuscript is appreciated. However, I still find a lack of clarity in the response to the previous review comments specifically in the technical aspects of the manuscript. 

To be specific, the authors should ultimately relate their kinetic analysis to some fundamental materials science.

The authors should also refer key publications in this aspect, and discuss their results accordingly.

The authors should improve the introduction and cite key references in the fundamental aspects of grain boundary pinning and grain growth in austenitic stainless steels. 

Author Response

Comment 1: the authors should ultimately relate their kinetic analysis to some fundamental materials science. The authors should also refer key publications in this aspect, and discuss their results accordingly.

According to your comment, we discuss the Q and n value.

Original

Revised

167

From the Table 3 result, it’s can   see that Mg content from 1.5 increase to 22 ppm, the Q exponent also   increase. But the Mg content are 5.6 and 11 ppm, the Q exponent is decrease. In general, the activation   energy Q for grain growth is affected by the amount and kind of the alloying   elements. When the materials has the inclusion pinning and the element of   solute drag. It’s make the grain growth difficult, the activation energy will   increase. It   means that there are some mechanism occur let the Q exponent decrease. And   the n exponent m   depends on the grain growth mechanism. The m exponent is 2, meaning that the   materials has no defects or inclusions. When the m exponent is 3, several   phenomena like inclusions in the grain are produced. It’s means that the   pinning effect will occur. The value is 4, the alloy element with diffusion   in the grain boundaries and produce the inclusion, meaning that the pinning   effect and the effect of solute drag will occur [55]. Thus, the effect of   pinning force and solute drag on austenite grain mobility were considering in   the next section.

Comment 2: The authors should improve the introduction and cite key references in the fundamental aspects of grain boundary pinning and grain growth in austenitic stainless steels.

Respond: According to your comment, we modify the introduction.

Original

Revised

167

According to previous   research, Zener pinning is the influence of a   dispersion of spherical particles [56], ellipsoidal particles [57] and cylindrical particles   [58] on the movement of grain boundaries. The   particles act to prevent the motion of grain boundaries by exerting a pinning   force which counteracts the driving force of grain boundaries. And the pinning efficiency against grain growth by particle is determined by   particle size [56].

Reviewer 3 Report

The authors amended the manuscript according to reviewers' report. This manuscript can be acceptable.

Author Response

Comment : The authors amended the manuscript according to reviewers' report. This manuscript can be acceptable.

Respond: 

Thank you for your suggestion.

Reviewer 5 Report

I appreciate the effort of correction of manuscript, but, unfortunately, cannot agree with author´s response.

1.)    Phase identification by means of OM-software is not sufficient since crystallographic structure cannot be detected by optical microscope. Please provide a proof based on crystallographic information (EBSD/XRD).

2.)    The argument that only the grain size of “normal grain growth” is considered cannot be accepted because of importance to analyze the whole microstructure in order to assess the influence of Mg on grain boundary mobility.

3.)    The “grain boundary" are missing in the title of the article: “The Influence of Mg-based Inclusions on the Grain Boundary Mobility of Austenite in SS400 Steel”

Author Response

Comment 1: Phase identification by means of OM-software is not sufficient since crystallographic structure cannot be detected by optical microscope. Please provide a proof based on crystallographic information (EBSD/XRD).

According to your comment, we do the XRD experiment to determine the crystallographic structure of martensite and austenite. But the austenite is the high temperature phase. It can not determine by XRD. Thus, we use the reference. to discuss it. Figure 1 shows the XRD experiment of martensite. Figure 2 shows the reference result of austenite. It can see that the martensite and austenite have different crystallographic structure.

Figure 1. XRD result of martensite.

Figure 2. XRD pattern of reference (γ is austenite, α is ferrite)

Reference: J. L. Garin, R. L. Mannheim, Rietvld Quantitative Analysis of Cast Super Duplex Steel. Power Diffraction. 2012, 27, 131-135.

Comment 2: The argument that only the grain size of “normal grain growth” is considered cannot be accepted because of importance to analyze the whole microstructure in order to assess the influence of Mg on grain boundary mobility.

Respond:

At the beginning, we consider is normal grain growth to calculate the Kinetic analysis of grain growth. If we consider is abnormal grain growth, the m exponent will be overrated. Thus, we choose the normal grain growth to calculate grain size.

Comment 3: The “grain boundary" are missing in the title of the article: “The Influence of Mg-based Inclusions on the Grain Boundary Mobility of Austenite in SS400 Steel”

Respond: thank you for your suggestion, the title was modified to correct.

Original

The Influence of Mg-based   Inclusions on the Mobility of Austenite in SS400 Steel

Revised

Title

The Influence of Mg-based Inclusions   on the Grain Boundary Mobility of Austenite in SS400 Steel

Round  3

Reviewer 1 Report

Previous review comments are addressed, However, the level of language used in the additional text is very poor and should be corrected.

Author Response

Comment: Previous review comments are addressed, However, the level of language used in the additional text is very poor and should be corrected.

Respond:

Thank you for your suggestion, we modify to the current one.

Original

L171

From   the Table 3 result, it’s can see that Mg content from 1.5 increase to 22 ppm,   the Q exponent also increase. But the Mg content are 5.6 and 11 ppm, the Q   exponent is decrease. In general, the activation energy Q for grain growth is   affected by the amount and kind of the alloying elements. When the materials   has the inclusion pinning and the element of solute drag. It’s make the grain   growth difficult, the activation energy will increase. It means that there   are some mechanism occur let the Q exponent decrease. And the n exponent m   depends on the grain growth mechanism. The m exponent is 2, meaning that the   materials has no defects or inclusions. When the m exponent is 3, several   phenomena like inclusions in the grain are produced. It’s means that the   pinning effect will occur. The value is 4, the alloy element with diffusion   in the grain boundaries and produce the inclusion, meaning that the pinning   effect and the effect of solute drag will occur. Thus, the effect of pinning   force and solute drag on austenite grain mobility were considering in the   next section.

Revised

L171

The   result in Table 3 shows that the increase in Mg content from 1.5 to 22 ppm   increased the Q exponent. However, the Q exponent decreased when the Mg   content was 5.6 and 11 ppm. In general, the activation energy Q for grain   growth is affected by the amount and type of alloying elements. For materials   with the inclusion pinning and element of solute drag, grain growth is   difficult and activation energy increases. In some mechanisms, the Q exponent   decreases. The m exponent depends on the grain growth mechanisms. When the m exponent is 2, the   materials have no defects or inclusions. When the m exponent is 3, several   phenomena, such as inclusions in the grain, are observed, which indicate the occurrence   of the pinning effect. When the m exponent is 4, the alloy element exhibits diffusion   in the grain boundaries and produces inclusion, which indicates the   occurrence of the pinning and solute drag effects [55]. Thus, the effects of the   pinning force and solute drag on austenite grain mobility are considered in   the subsequent section.

Reviewer 5 Report

Dear authors, thank you for comments and for improvement of manuscript.

Author Response

Comment : Dear authors, thank you for comments and for improvement of manuscript.

Respond: Dear reviewer, thank you for your suggestion